# Effect of Transgenesis on mRNA and miRNA Profiles in Cucumber Fruits Expressing *Thaumatin II*

**DOI:** 10.3390/genes11030334

**Published:** 2020-03-20

**Authors:** Magdalena Ewa Pawełkowicz, Agnieszka Skarzyńska, Małgorzata Sroka, Maria Szwacka, Tomasz Pniewski, Wojciech Pląder

**Affiliations:** 1Department of Plant Genetics, Breeding & Biotechnology, Institute of Biology, Warsaw University of Life Sciences, Nowoursynowska Street 159, 02-776 Warsaw, Poland; 2Institute of Plant Genetics, Polish Academy of Sciences, Strzeszyńska Street 34, 60-479 Poznan, Poland

**Keywords:** transgenic cucumber, thaumatin, RNA-seq, miRNA-seq, transcriptome comparison, unintended effects

## Abstract

Transgenic plants are commonly used in breeding programs because of the various features that can be introduced. However, unintended effects caused by genetic transformation are still a topic of concern. This makes research on the nutritional safety of transgenic crop plants extremely interesting. Cucumber (*Cucumis sativus* L.) is a crop that is grown worldwide. The aim of this study was to identify and characterize differentially expressed genes and regulatory miRNAs in transgenic cucumber fruits that contain the *thaumatin II* gene, which encodes the sweet-tasting protein thaumatin II, by NGS sequencing. We compared the fruit transcriptomes and miRNomes of three transgenic cucumber lines with wild-type cucumber. In total, we found 47 differentially expressed genes between control and all three transgenic lines. We performed the bioinformatic functional analysis and gene ontology classification. We also identified 12 differentially regulated miRNAs, from which three can influence the two targets (assigned as DEGs) in one of the studied transgenic lines (line 224). We found that the transformation of cucumber with *thaumatin II* and expression of the transgene had minimal impact on gene expression and epigenetic regulation by miRNA, in the cucumber fruits.

## 1. Introduction

Cucumber is a plant of great economic importance. It has been widely bred using modern biotechnology methods to create varieties with new features. Work on the transformation of cucumber has been underway at the Department of Plant Genetics, Breeding and Biotechnology (Warsaw University of Life Sciences, Poland) since 1996, when transgenic cucumber plants with the *thaumatin II* gene were obtained [1].

Thaumatin is a protein derived from *Thaumatococcus daniellii* (Benn.) Benth., which is a monocotyledon that grows naturally in tropical forests of West Africa. The fruit is pyramidal shaped and is filled with a colorless gel that contains thaumatin [2]. The thaumatin family contains several closely related proteins, including thaumatin I, thaumatin II, and thaumatins III a–c. Thaumatin is about 100,000 times sweeter than sucrose. It has a high affinity for the receptors responsible for the sweet taste sensation, which makes its perception much higher than that for sugars [3]. As thaumatin II is the only sweet-tasting protein in its family, it is assumed that the sweet taste is due to the modification of five lysine residues in this protein [4]. In addition, thaumatin is non-caloric, non-toxic, and is accepted by nutrition authorities [2]. 

The thaumatin family proteins are involved in the differentiation of the flower [5] and in fruit maturation [6]. Members of this family have been classified as pathogen-related 5 (PR-5), so they are involved in immune responses during the fungal attack or in defense reactions to abiotic environmental stresses such as drought, low temperature, or salinity [7]. 

In addition to their antifungal activity, several pathogen-related proteins have the capacity to bind glucans and possess glucanase activity [8]. These proteins can interfere with the fungal cell wall, which probably explains their ability to fight fungal infections [8].

Thaumatin is used as a sweetener and flavor enhancer in the food industry and as a component of special low-calorie diets in pharmaceutics [9]. It gives a sweet taste to food and medicinal products in very low concentrations (of the order of 10^−8^ M), so it does not significantly change the weight of the product [10]. The production of thaumatin from the fruits of wild *T. daniellii* plants is limited [9] and is not enough to meet the market demand. It, therefore, seems reasonable to obtain thaumatin using transgenic crops. The transformation has the advantage of being able to produce relatively cheap raw materials in large quantities. To date, *thaumatin II* has been successfully expressed in potato [11], cucumber [12], tomato [13,14], pear [15], carrot [16], strawberry [17], and apple trees [18], as well as other species of edible plants. The fruit and vegetative organs of the transgenic plants had a sweet taste, indicating *thaumatin II* was expressed and the thaumatin II protein was correctly transcribed.

In a previous study, cucumber transformation was carried out using a disarmed *Agrobacterium tumefaciens* strain with binary plasmid pRUR528 containing the *thaumatin II* cDNA [1]. The obtained plants had one, two, or five transgene integration sites [12,19] and were characterized in subsequent generations after a comprehensive evaluation of the transgenic lines [10]. The T1 generation was obtained by self-pollination, followed by the T2 generation in which the level of transgene expression was related to an increase in organoleptically confirmed sweetness in fruits [12]. 

In addition to the intended change obtained by the introduction of *thaumatin II* to the cucumber genome (i.e., the sweet taste of the fruit), unintended effects (unrelated to the target traits) [20] also have been observed. They include increased emission of volatile aroma compounds by cucumber fruits, which give them a more intense flavor [21], as well as increased tolerance to downy mildew [22]. Previous analysis of transgenic cucumber plants showed changes in the morphological and anatomical structures of the leaves, such as the wax thickness and secondary epidermal cell wall, and the arrangement of mesophyll cells. Such changes may provide a more effective mechanical barrier against the penetration of a pathogen [12,22]. A nutritional safety assessment of transgenic cucumbers carried out on rats found no negative effects on the health of the tested animals [23]. 

In this study, we investigated the fruits of transgenic cucumbers with the *thaumatin II* gene to detect transcriptomic changes that may indicate unintended effects at the molecular level and show the influence of the transgene on the other genes in the commercial product which is cucumber fruit. Referring to this, we checked the changes in miRNA profiles. This assay screened for differentially expressed genes (DEGs) and found miRNAs providing a new theoretical basis for studies investigating the effects of transgenesis on epitranscriptome and further possible changes in signaling network in cucumber.

## 2. Materials and Methods

### 2.1. Plant Material

Four cucumber lines, three transgenic cucumber lines containing the *thaumatin II* gene (212, 224, and 225) and one control line (B10 line), were used for the RNA sequencing (RNA-seq) analysis. The transgenic plants were obtained by transformation using the *A. tumefaciens* strain that contained a binary plasmid with *thaumatin II* cDNA driven by the CaMV 35S promoter, together with a reporter construct consisting of the nopaline synthase (nos) promoter driving the expression of the *neomycin phosphotransferase II* (*nptII*) coding sequence [1]. To develop lines with stable transgene expression, after transformation, under strictly controlled and specific greenhouse conditions (25–27 °C day/18–20 °C night, with 16 h photoperiod and light intensity 1500 μmol·m^−2^·s^−1^ [12]), we conducted manual self-pollination until the T9 generation. Plants (T9) were cultivated in the field in a mesh tunnel with irrigation for two months (season 2014). The field experiment was carried out using the random blocks method. Ten plants per each line were seeded and phenotypically assessed. Two weeks after self-pollination, fruits were sliced and harvested into the liquid nitrogen. Replicas in the experiment were single sliced tissue from pulp fruits from three different plants per line. The highly inbred cucumber B10 line was used as the control. Three biological replicates were used for sequencing and real-time quantitative RT-PCR (qPCR) validation.

### 2.2. DNA and RNA Extraction

Total DNA was extracted from 100 mg of tissue using a DNeasy Plant Mini Kit (Qiagen, Hilden, Germany) according to the manufacturer’s protocol. Total RNA was extracted using a RNeasy Mini Kit (Qiagen, Germantown, USA). The nucleic acid concentration and quality were measured using a NanoDrop 2000 Spectrophotometer (Thermo Fisher Scientific, Waltham, MA, USA). The concentration of pure RNA was adjusted to 100 ng·µL^−1^. RNA integrity (RIN) was assessed using a Bioanalyzer 2100 (Agilent, Santa Clara, CA, USA) and the RNA 6000 kit according to the manufacturer’s instructions. The RNA samples used for sequencing had total amounts of RNA ≥22 µg, concentrations ≥500 ng·µL^−1^, major ribosomal subunit ratio 28S:18S ≥1, and RIN ≥8. 

For the qPCR, RNA was treated with DNaseI (TURBO DNA-free kit, Ambion, Austin, TX, USA) and checked by PCR for lack of DNA contamination. cDNA was synthesized using a High-Capacity cDNA Reverse Transcription Kit (Thermo Fisher Scientific) according to manufacturer’s instructions.

### 2.3. Confirmation of Transgenicity

Insertion of the transgene expression cassette into the cucumber genome was confirmed by PCR amplification using primers designed to amplify the *thaumatin II* and *nptII* (kanamycin resistance) gene fragments. Two pairs of primers designed to amplify the *thaumatin II* gene fragment were used: Tha_S_F 5′-TCCTCCTCCTCACGCTCTCC-3′ and Tha_S_R 5′-AGCCAATCCCCACACACATA-3′ [12]; and Tha_M_F 5′-GCCACCTTCGAGATCGTCAAC-3′ and Tha_M_R 5′-TGTCGTCGAAATAGC AGTCG-3′ [24]. Primers nptII_F 5′-GAGGCTATTCGGCTATGAACTG-3′ and nptII_R 5′-ATCGGGA GCGGCGATACCGTA-3′ were used to amplify the *nptII* gene fragment. Primers designed to amplify the reference gene of ubiquitin extension protein were used as the positive control UBIep_F 5′-CACCAAGCCCAAGAAGATC-3′ and UBIep_R 5′-TAAACCTAATCACCACCAGC-3′. The non-transformed plant genomic DNA was used as the control. PCRs were carried out in a 25 µL reaction mixture containing 100 ng of template DNA using DreamTaq polymerase (Thermo Fisher Scientific) according to the manufacturer’s protocol. The PCR conditions were as follows: 95 °C for 5 min; 35 cycles of 95 °C for 30 s, 62 °C for 30 s, 72 °C for 50 s; and a final extension at 72 °C for 10 min.

### 2.4. Library Construction and Sequencing

Approximately 5–10 μg of total RNA was used to construct each RNA-seq and miRNA-seq libraries. The RNA samples used for the miRNA-seq were the same as those used for the mRNA-seq. Polyadenylated RNA purification, RNA fragmentation, cDNA synthesis, and PCR amplification was performed according to the Illumina RNA-seq protocol (Illumina, Inc., San Diego, CA, USA). For miRNA library TrueSeq Small RNA library Kit (Illumina, Inc.) was used. Parallel sequencing was performed on an Illumina HiSeq 2000 platform (McGill University Genome Quebec Innovation Centre, Montreal, QC, Canada). Three replicates were used for each transgenic line and for the control B10 line. We obtained 100-bp paired-end sequence reads for RNA-seq and 50-bp single-end sequenced reads for miRNA-seq. The read quality was evaluated using the Illumina purity filter, percent low quality reads, and distribution of phred-like scores at each cycle. All the presented data passed the quality control filtering on the basis of these metrics. FastQC [25] was used to assess the quality of the short reads. The sequences generated in this study have been deposited in the Sequence Read Archive (SRA) at the National Center for Biotechnology Information under accession numbers BioProjectsPRJNA572579, PRJNA578623, PRNJA610511 and PRJNA610495.

### 2.5. Bioinformatic Analysis

The *C. sativus* genome sequence B10v3 (GenBank: LKUO00000000) and annotated genes were retrieved from PCC Genomics (Polish Consortium of Cucumber Genome Sequencing) [26]). The RNA-seq reads from this project, together with transcriptomic data from previous PCC Genomics research, were used to generate the *de novo* transcriptome assembly [26]. Gene expression was analyzed by comparing the transcriptome data between the transgenic and wild-type plants. Gene expression values were estimated using the Salmon software package [27] with sequence-specific and GC content bias enabled. Differential expression analysis was performed using the limma package [28] following a previously published protocol [29]. Differentially expressed genes (DEGs) also were detected using DESeq [30]. Genes were considered to be differentially expressed when the FDR (false discovery rate, adjusted p-value) was <0.001 and the fold change (FC) was >1.5. Clustering analysis was performed using MeV 4.9.0 [31] with the Pearson correlation. Functional annotation and Gene Ontology (GO) classification of the DEGs were carried out using Blast2GO software [32]. To detect domains in the translated protein sequences, we used Pfam [33], InterProScan [34], and Prosite [35]. To predict the location of the proteins in cells, we used TargetP [36] and WoLF PSORT II [37] and also checked if they might be transmembrane proteins using TmHMM [38]. To identify the known miRNAs, the clean reads were used in the BLAST search against known mature miRNAs and pre-miRNAs of miRBase (Version 21.0) [39]. The expression quantity was calculated according to the transcripts per million (TPM). Differential expression analysis was performed with DESeq2 [40] using default parameters. The targets of the mature miRNA sequences were identified using psRNAtarget [41]. The STRING algorithm (version 10.5) [42], with *A. thaliana* as a model, was applied for additional analysis of the possible interaction between the differentially expressed targets for miRNAs. We used PlantCARE [43] to detect cis-acting elements in the promoter regions (1000-bp long upstream sequences from the start codon of the predicted genes).

### 2.6. Validation of DEGs Expression Profiles by qPCR

We designed qPCR primers for specific genes using Primer3 (version 2.3.6). Reference gene UBIep was chosen based on the experimental results as previously described by Skarzyńska et al. [33,44]. Details of the qPCR primers and amplicon lengths are provided in Appendix A. To verify the RNA-seq data and compare transcript levels among the analyzed cucumber lines, we selected 16 DEGs for verification by qPCR. All qPCRs were completed using three biological replicates, with three technical replicates. The cDNA used for the qPCRs was reverse transcribed from 1 µg total RNA and diluted (1:5). The qPCRs were completed with 4 µL cDNA and the Power SYBR® Green PCR Master Mix (Thermo Fisher Scientific) on an Applied Biosystems 7500 Real-Time PCR System (Thermo Fisher Scientific). The PCR program was 50 °C for 2 min, 95 °C for 10 min, followed by 40 cycles of 95 °C for 15 s and 60 °C for 1 min. A melting curve analysis was completed immediately after the qPCR. The amplification efficiency for all primers was 107%–110%. The mean amplification efficiency was assessed using LinRegPCR (version 2015.3) [34,45]. Relative expression levels were determined according to the 2^−ΔΔCt^ method with Rstudio and EasyqPCR from the Bioconductor software package [46]. 

## 3. Results

We sequenced the transcriptomes from small fruits (two-weeks after pollination) of three transgenic cucumber lines (212, 224, and 225) and their wild-type B10 line to detect differentially expressed genes (DEGs) and to check whether unintended effects of transformation may occur at the molecular level in cucumber. The aim of this study was to determine if the *thaumatin II* transgene could change the transcription profiles of other genes, (directly or through epigenetic regulation by miRNA), thus reprogramming native molecular networks that could influence on fruits yield and quality.

### 3.1. Confirmation of Transgenes Stability and Expression

To confirm the true transgenic lines, genomic DNA was isolated from T9 progenies of the cucumber transgenic plants from the three lines for PCR analysis using primers specific for *thaumatin II* and *nptII*. The PCR analysis, performed on DNA, showed that the expected 815-bp and 182-bp fragments of *thaumatin II* and 110-bp fragment *nptII* gene were amplified in all transgenic samples (Figure 1a). No amplified PCR product was detected in the control of B10 plants. The qPCR analysis showed the expression on mRNA level of the transgene transcripts in fruits in all tested transgenic lines (Figure 1b,c); no amplification was observed from the cDNA of the control plants. The relative expression levels of *thaumatin II* were 1.969, 1.963, and 0.317 in 212, 224, and 225 lines, respectively. The relative expression levels of *nptII* were 2.524, 1.888, and 0.0563 in 212, 224, and 225 lines, respectively.

### 3.2. Identification of DEGs

We obtained 42.9–53 million reads from the libraries of the analyzed lines. The mean duplication was 32%, the average Phred quality was 35, and the mapping quality was 92% (Appendix A). Comparison of the fruit transcriptome profiles of lines 212, 224, and 225 with the fruit transcriptome profiles of the wild-type B10 cucumber line revealed three (two down- and one up-regulated), 38 (eight down- and 30 up-regulated), and six genes (all up-regulated) that were differentially expressed in 212, 224, and 225 lines, respectively (Figure 2a). Detailed quantity distributions of the DEGs are presented in Figure 2b. The DEG expression profiles were consistent for each replicate but were clearly differentiated in the DEG expression profiles between the three transgenic lines (Figure 2c). The overlapping analysis revealed two DEGs that were common among the transgenic lines: *G7760* (up-regulated in lines: 224 and 225) and *G14398* (up-regulated in lines: 212 and 225).

### 3.3. DEGs in the Transgenic 212 Line

We detected only three DEGs in 212 line; two were down-regulated (*G20340* and *G4326*) and one was up-regulated (*G14398*) (Table 1). One of the down-regulated genes (*G4326*) encoded a homolog of the membrane-associated prohibitin (PHB) domain protein. The Pfam analysis indicated that the encoded protein can bind to lipids and can assemble into membrane-bound oligomers involved in the ubiquitination process. The functions of the other two genes (*G20340* and *G14398*) are not known. 

### 3.4. DEGs in the Transgenic 224 Line

We detected 38 DEGs in 224 line; eight were down-regulated and 30 were up-regulated (Table 1). The function of three of the genes was uncharacterized (*G3896, G16954* and *G17428*) and two of the genes encoded lincRNAs (*G853* and *G1149*). Eight genes were related to defense and resistance against pathogens (*G11114*, *G9940*, *G12465*, *G6838*, *G15513*, *G19560*, *G17693*, *G12648*, and *G149*) and five were associated with lipid metabolism (*G19558*, *G9998*, *G21621*, *G12174*, and *G14791*). The rest genes encoded proteins connected with membranes (*G10813* and *G6726*), cell walls (*G17313* and *G1817*), oxidation (*G14363* and *G7760*), ions (*G10467* and *G6441*), transcriptional factors (*G12598* and *G3425*), chloroplast (*G15599*), transferase (*G13243*), ribosome (*G14592*), hormones (*G7198*), ribonuclease (*G8549*), abiotic stress (*G828*) and light (*G6936*). 

### 3.5. DEGs in the Transgenic 225 Line

We detected six DEGs in 225 line, all of which were up-regulated. Three of them encoded lincRNAs (*G10371*, *G15468*, and *G20381*) and one encoded an uncharacterized protein (*G14398*), so their function could not be determined. One gene (*G6182*) encoded a protein with homology to calcium- calmodulin-dependent serine kinase, so it may play a role in signal transduction pathways that involve calcium as a second messenger and ion binding. The other gene (*G7760*) encoded a protein that was similar to the pentatricopeptide repeat-containing protein that regulates oxidative respiration and environmental responses in plants.

### 3.6. Confirmation of Illumina RNA-seq Expression by qPCR

To validate the DEG expression levels calculated from the RNA-seq data, we randomly selected 15 DEGs for verification by qPCR (Appendix A) as follows: two genes (*G20340*, *G4326*) for 212 line; 11 genes (*G17693*, *G13243*, *G10813*, *G10467*, *G12598*, *G6726*, *G19560*, *G1114*, *G14363*, *G19558*, *G7760*) for 224 line; and two genes (*G7760*, *G15468*) for 225 line. A highly reliable reference gene (*CsUBIep*) was used to normalize the qPCR data according to the 2^−ΔΔCt^ method. The gene expression levels obtained by qPCR and from the RNA-seq data were positively correlated for 14 (93%) of the analyzed DEGs (Figure 3); the exception was *G20340*. This inconsistency may be explained by the different sensitivities of the two methods and algorithms.

### 3.7. Functional Annotation of the DEGs

The DEGs were annotated using BLAST and functionally categorized with GO terms using Blast2GO software. We used a set of plant-specific GOSlim terms and clustered them under the three main GO categories: biological process, molecular function, and cellular component (Figure 4). 

The most abundant GOSlim terms under biological process were: response to stimuli (biotic and abiotic), metabolic processes, and biosynthetic processes in 224 line and protein modification and signal transduction for 225 line. The most frequent GOSlim terms under molecular function were connected with binding group hydrolase and catalytic activity in 224 line: nucleotide and protein binding, also kinase activity in 225 line, and protein binding in 212 line. The most frequent GOSlim terms under cellular components were membrane, extracellular region and cell wall in line 224, membrane group in 212 line, and membranes, nucleus and cytoplasm in 225 line. The GOSlim terms are presented in Appendix A. We predicted the location of specific proteins in the cells using the WolfPsort and TargetP programs. TargetP connected eight proteins with mitochondria, seven with chloroplast, and two with other plastids. WolfPsort, which has a more extensive database, predicted seven protein connected with chloroplast and ten with other plastids, seven with nucleus, six with cytosol, three with mitochondria, two with vacuoles, two with extracellular space, one with endoplasmic reticulum, for seven there were no predictions (Appendix A). SignalP predicted that six proteins had signal peptides and TmHMM predicted that two proteins, those encoded by *G13243* and *G6726*, had transmembrane helixes (Appendix A).

### 3.8. Identification of Differentially Expressed miRNA

We have identified altered expression of 12 miRNA molecules (Table 2, Appendix A) in three analyzed transgenic lines. There were: one miR320 in 212 line which was down-regulated, eight differentially expressed miRNAs in 224 line (four down-regulated: miR27, miR131, miR218, miR222 and five up-regulated: miR42, miR93, miR123, miR206, miR289), two miRNA in 225 line (down-regulated miR218 and up-regulated miR206). Among predicted targets presented (Appendix A) only two targets in 224 line showed differential expression: *G10467* (ion binding-calcium, annexin) and *G12598* (NF-Y nuclear transcription factor Y subunit). The rest targets did not reveal significant changes in expression in fruit transcriptomes (Figure 5).

### 3.9. Bioinformatics Analysis of the Upstream Promoter Regions of the DEGs

We detected 2496 motifs (an average of 56.73 motifs per gene) in the promoter regions of the DEGs using the PlantCARE program (Appendix A). The highest number of motifs in one promoter region was 96 and the lowest was 16. Details of the predicted motifs are presented in Appendix A. The most common groups of motifs in all lines were TATA-box and CAAT-box (Figure 6, Appendix A). Common motifs in three lines were also ATTATATA-box, MYC, MYB and ARE. The highest number of the motif was found in 224 line as this line had the highest number of DEGs. Very important site: CCAAT (which is also included in CAATT-box) has appeared 58 times in 24 genes (Appendix A) in 224 line. The CAAT-box has appeared 531 in 37 of DEGs (Appendix A). 

The occurrence of detected *cis*-acting elements in the promoters of the DEGs is shown in Figure 7 (Appendix A). The highest number of motifs was associated with conserved *cis*-acting elements (core promoters transcription site and enhancers) and light response.

## 4. Discussion

In this study, we confirmed the genomic stability and expression of the *thaumatin II* gene in the T9 generation in three transgenic lines (212, 224, and 225). The results showed that *thaumatin II* was expressed in all three lines, but lines 212 and 224 had about 6.7-fold higher *thaumatin II* expression levels than line 225. This is consistent with an earlier study of transgenic cucumber lines in which the expression of the thaumatin II transgene was confirmed in fruits [47]. By examining the subsequent generation of transgenic plants and confirming the expression of the *thaumatin II* transgene, we confirmed that it was stably integrated into the cucumber genome. The different expression levels of the transgene among the lines may be because of different localization of integration sites in the genome and/or the influence of chromatin modification and chromatin-mediated transgene regulation as reported by Kohli et al. [48]. The transgenic cucumber plants have been subjected to a series of tests in successive generations aimed at a comprehensive evaluation of transgenic lines, as reviewed by Szwacka et al. [10]. 

In this study, we analyzed the transcriptomes of cucumber fruits using RNA-seq technology and performed an extensive in silico study of the DEGs to find how transgene integration altered the transcriptome profiles. 

In 212 line, the expression of *G4326*, which encoded the membrane-associated PHD domain protein involved in the ubiquitin pathway and thus is responsible for the degradation of proteins, was significantly reduced. Stopping the ubiquitination process causes dysfunctions of proteins and can lead to the accumulation of poorly folded proteins and the formation of aggregates, which disturbs the functioning of cells [49]. Thus, in 212 line, the ubiquitination process, especially with regard to the membrane-associated proteins, may be disrupted.

The largest number of DEGs (38) was found for 224 line. They were associated with many cellular processes, including lipid metabolism, oxidation, growth, transcription (transcriptional factors), and plant defense as well as processes related to light, hormones, ion homeostasis, cell walls and membranes. One of the DEGs (*G11114*) encoded the chaperone DnaJ. Wang et al. [50] have shown that the chloroplast-localized *DnaJ* gene of tomato (*LeCDJ2*) was involved in resistance to *Pseudomonas solanacearum* in transgenic tobacco. In tobacco, the DnaJ protein may play a role in plant immunity to virus infection [51,52]. Another DEG (*G9940*) identified in 224 line encoded the defense-related major latex protein (MLP). In cotton, the expression of MLP was induced by inoculation with the fungus *Verticillium dahlia*, indicating its expression was a response to defense signaling molecules [53]. Knockdown transgenesis experiments revealed increased susceptibility of cotton plants to *V. dahlia* infection and, in tobacco, overexpression of *GhMLP28* influenced the disease tolerance of transgenic plants [53]. Plants recognize pathogen-associated molecular patterns and use extracellular leucine-rich repeat receptors for elicitor recognition, serine/threonine kinases (encoded by *G12465*) for downstream processing, and mitogen-activated protein kinases for the signal relay. Cytoplasmic serine/threonine kinase conferred race-specific resistance to *Pseudomonas syringae* by recognizing the effector protein avrPTO. PTO does not contain a known ligand-binding motif but is involved in both elicitor recognition and phosphorylation [54]. The O-glucosyltransferase 1-like protein encoded by *G15513* belongs to the glycosyltransferase family and may be involved in the response to fungal infection. In *Arabidopsis thaliana*, members of this family were shown to display distinct induction profiles, indicating potential roles in stress or defense responses, notably for infection with *P. syringae* [55]. The differentially expressed *G19560* gene encoded β-amyrin, a protein of unknown function, which is an intermediate in the synthesis of more complex triterpene glycosides associated with plant defense [56]. Based on the gene ontology annotation it could be concluded that changed expression of some genes may be associated with increased tolerance to pathogens and this could be correlated with the introduction of *thaumatin II*. In previous analysis it was described that 224 line possess increased tolerance to downy mildew [12,22]. This is in line with other analyses of transgenic plants with *thaumatin II* transgene and enhancement of pathogen resistance [17,57].

Chalcone synthase (CHS), encoded by *G17693*, is a key enzyme of the flavonoid/isoflavonoid biosynthesis pathway. Besides being part of the plant developmental process, CHS expression is induced in plants under biotic stresses, such as bacterial or fungal infection. The CHS protein can help plants produce more flavonoids, isoflavonoid-type phytoalexins, and other related metabolites to protect against pathogens [58]. The Kaschani group has studied the expression of the gene encoding serine hydrolase in plants infected with the fungal pathogen *Botrytis* sp. They found a correlation between the level of serine hydrolase and the degree of *Botrytis* infection [59]. One of the DEGs (*G12648*) in our study encoded serine hydrolase and was up-regulated in 224 line (Table 1), indicating serine hydrolase may play a role in response to pathogen infection in 224 line.

The DEG G149 encoded germacrene A synthase, which is a key cytosolic enzyme in the sesquiterpene lactone biosynthesis pathway. Sesquiterpene emission from leaves of several plant species has been shown to play important roles in direct and indirect chemical defense against pathogens [60]. Notably, we found an increase in *G149* expression in 224 line (Table 1), implying this gene may be associated with increased tolerance to pathogen attack in this line.

Lipids and lipid metabolites are important elements in the response of plants to biotic stresses and also are involved in plant–microbe interactions. Despite the extensive studies of lipases in lipid homeostasis, the involvement of lipases in plant immunity remains largely unknown. In this study, we identified many DEGs connected with lipid metabolism (*G19558*, *G9998*, *G21621*, *G12174*, and *G14791*). In particular, the GDSL esterases/lipases encoded by *G14791* are characterized by the conserved GDSL motif and belong to a subfamily of lipolytic enzymes with broad substrate specificity. In rice, GDSL lipases were shown to function in the plant’s immune responses [61]. Further, Gao et al. [61] showed that *OsGLIP1* and *OsGLIP2* negatively regulated rice defense by modulating lipid metabolism, thus providing new insights into the function of lipids in plant immunity. Genes related to lipid biosynthesis may be associated with resistance to downy mildew because the plant can produce a thicker cutin layer, which will increase the mechanical resistance of the plant to the pathogen, as was discussed in transgenic cucumber by Szwacka et al. [22].

Other DEGs connected with the cell wall (*G17313* and *G1817*) and membranes (*G10813* and *G6726*) were detected in 224 line (Table 1). Resistance to penetration at the epidermis is a key component of basal defense against disease and depends on the cell wall at the site of pathogen penetration. Hardham et al. [62] found that the formation of cell walls and membranes are crucial to the first steps of infection because they are the first mechanical barriers of the plant defense system. At the site of infection, many rapid changes take place in actin microfilaments, actin-dependent transport of secretory products to the infection site, and activation of callose synthesis. These changes allow the plant to recognize the infection process and to detect the presence of pathogens on their surface by perceiving both chemical and physical signals of pathogen origin, which triggers a whole cascade of events leading to defensive reactions [62].

The protein encoded by *G10467* was annexin 3, which is responsible for the binding of calcium ions. Annexins are a family of calcium- and phospholipid-binding proteins that are present in all plant species. Members of this family are involved in many important cellular processes. The abundance levels of these proteins change after plants are treated with light of various intensity, as well as after exposure to biotic and abiotic stresses [63]. 

The other DEGs in 224 line are associated with many different processes, including oxidation, ion binding, and transcriptional factors, and also are related to response to light and hormones. Thus, the introduction of *thaumatin II* into the genome of line 224 resulted in a significant increase in the expression of genes involved in various metabolic pathways and responses to external stimuli. However, this did not cause significant alterations in the growth and development of the plant’s morphology [22]. 

In 225 line, six DEGs were detected. Three of them (*G10371*, *G15468*, and *G20381*) were lincRNAs and one (*G14398*) was of unknown function. The up-regulated *G7760* encoded a protein that contained the pentatricopeptide motif that is responsible for protein binding. Over 450 of such proteins have been found in plants where they act specifically in mitochondria and plastids and are involved in RNA modification and splicing, as well as translation initiation and regulation [64]. The other up-regulated gene *G6182* encoded calcium- calmodulin-dependent serine kinase, which, according to Hsueh et al. [65], is a membrane-associated protein kinase that mediates processes primarily in organelles but also in the nucleus. Both of these encoded proteins are connected with organelles and may be partially dependent on one another in 225 line.

Most of the DEGs in the analyzed transgenic lines (212, 224, and 225) were assigned GO terms related to metabolic and cellular processes. Some DEGs were associated with response to stimuli, which is consistent with the idea that the increased resistance of transgenic plants to pathogens is related to the introduction of *thaumatin II* [22]. Under the molecular function category, most of the DEGs were assigned terms related to catalytic activity and binding of carbohydrates, small molecules and vitamins, and nucleic acids. Under the cellular component, the most enriched terms were cell or its components, and cell membrane.

Our analysis of the DEG promoter regions indicated, that most of the promoters contained motifs related to response to light. Light is one of the main stress factors associated with changes in gene expression in non-transformed plants [66]. Thus, the change in the expression levels of light-dependent genes may not be due to the presence of the transgene, but simply the plant’s response to the growth conditions.

The most frequently occurring motifs in the promoter sequences were CAAT-box, A-box, and TATA-box, which are commonly located close to the transcription start sites of genes [67]. Other cis-acting elements were transcription regulators, which are involved in many functions including flower development, cell differentiation, secondary metabolic changes, responses to biotic and abiotic stresses, and response to the hormones abscisic acid and ethylene.

There are three basic mechanisms of epigenetic genome control: DNA methylation, histone modification and gene expression controlled by small RNA including micro RNA [68]. In this study we performed an analysis of differentially regulated miRNA. In each transgenic line we identified small changes in miRNA profiles, however only in 224 line, changes in miRNA significantly influence the expression of their target genes (miR31: target *G10467* and miR218 together with miR222 – target: *G12598*). The type of correlation of expression between miRNA and their target molecules is worth the attention. These three miRNAs were down-regulated in 224 line. The low level of miRNAs probably influences the higher level of expression *G10467* and *G12598* (as confirmed by qPCR). The lower level of miR218 in 225 line did not change the expression of *G12598*. Based on the function of the *G12598* which is a TF histone core like (NF-Y) it can be expected that a change in NF-Y expression can have a significant impact on the expression of other genes. The NF-Y is a transcription factor that plays a role in complex together with subunit NF-YB and NF-YC (Appendix A), because individual subunits cannot regulate transcription independently but must form the complexes. 

The NF-YA subunit recognizes the CCAAT motif (CAAT-box) in promoters what allows for specific and stable DNA binding. The subunits NF-YB/NF-YC interacts with DNA via a non-specific histone fold domain-DNA (HFD-DNA) domain [69]. The TF NF-Y complex is recognized as a histone-like protein because the tridimensional structure can bind to DNA. This structure is similar to that of core histones H2A/H2B with the difference that H2A/H2B binds non-specific to DNA. By their similarity to histones, NF-Y probably affects chromatin remodeling and significantly influence gene expression [70]. Different combinations of subunits A, B and C can lead to a wide variety of NF-Y complexes which play a different role in various developmental stages, tissues and growth conditions [71]. Thus, we suppose that the higher number of DEGs in 224 lines, compared to 212 and 225 lines could be caused by miR218 a miR222 regulation by up-regulation of target NF-Y (*G12598*). The NF-Y could act with genes that possess CCAAT motif in promotor region and among DEGs, we detected such kind of motif in 24 genes (37 genes which possess CAAT-box). This may indicate strong interactions with the target *G12598* NF-Y sub-A and other DEGs in 224 line. We conclude that there is an inner epitranscriptomic regulation of genes expression by miRNA caused by transgenesis. 

The reduction in expression of the miR131 molecule on the *G10467* target gene seems to have less effect in the regulation of other genes than the miR218 and miR222 molecules on target *G12598*, due NF-Y is a multifunctional transcription factor. As mentioned previously, *G10467* encodes annexin3 and is strongly correlated with calcium ion binding and could play as the sensor, skeletal regulator, ion channels regulator, and proteins involved in membrane dynamics, actin modeling, Ca^2+^ ion movement and ROS regulation [72] (Appendix A).

## 5. Conclusions

The aim of this work was to detect any unintended effects of genetic transformation and possible changes in biological processes in plants with a transgene by the molecular analysis of three cucumber transgenic lines with the *thaumatin II* gene. Transgene stability was confirmed at both DNA and RNA levels, and the relative expression level of *thaumatin II* was measured. The RNA sequencing data were verified by qPCR analysis. The RNA-seq data analysis detected 47 DEGs and 12 miRNAs in three transgenic lines. The DEGs were functionally annotated with GO terms, which showed that most of them were involved in metabolic and biological processes associated with binding of ions, proteins, and nucleic acids, as well as with catalytic activity. Several generations after the plant transformation still maintains the proper transgene expression. Transgene integration could influence epigenome by changing the miRNA profiles, but the effects are minimal and mostly depend on the function of target mRNA molecules. 

## Figures and Tables

**Figure 1 genes-11-00334-f001:**
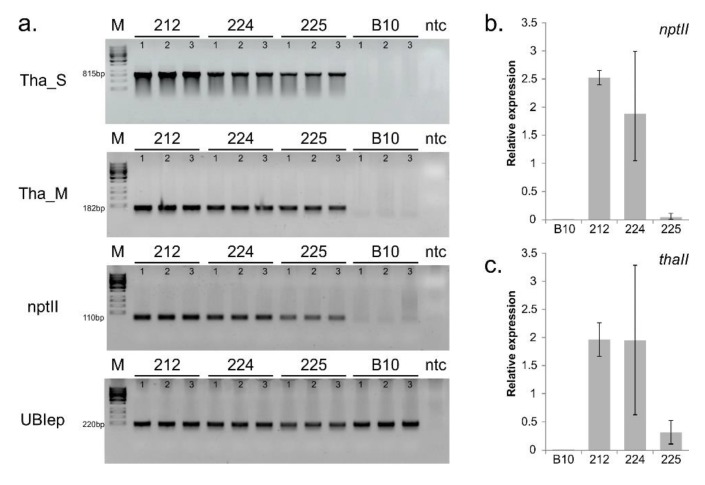
Confirmation of transgenicity. (**a**) PCR amplification of the *thaumatin II* and *nptII* transgenes in three transgenic cucumber lines (212, 224, and 225). Lane M: DNA molecular marker; Lanes 1, 2, and 3: individuals of 212, 224, and 225 lines; Lane B10: non-transgenic control plant; Controls: ntc, negative technical control; UBIep, reference gene; (**b**,**c**) qPCR analysis of mRNA transcripts in fruits of transgenic cucumber plants. Transgenic lines 212, 224, and 225 were compared with the wild-type B10 line; (**b**) expression of *nptII*, (**c**) expression of *thaumatin II*.

**Figure 2 genes-11-00334-f002:**
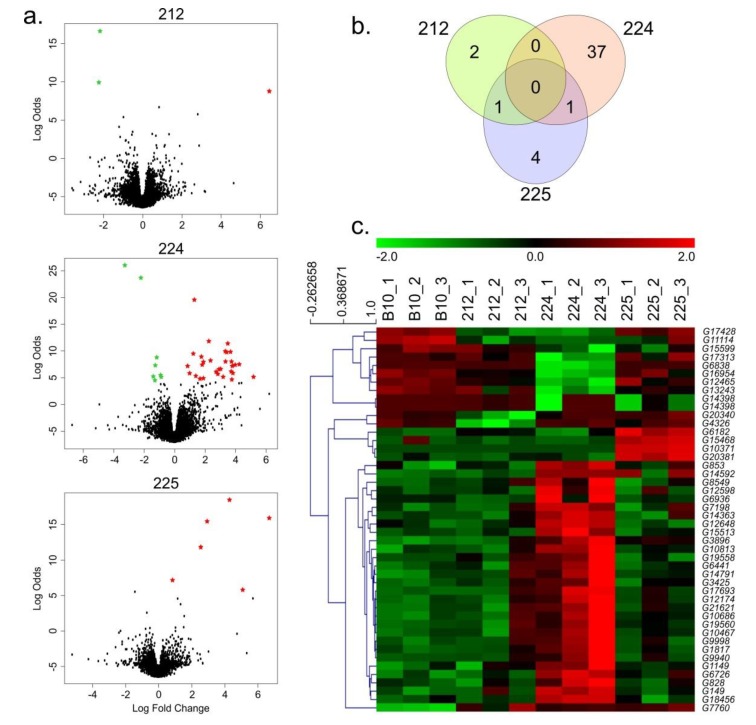
(**a**) Volcano plots of differentially expressed genes from three cucumber transgenic lines (212, 224, and 225) versus the wild-type B10 line. Green and red asterisks indicate significant results; black dots indicate non-significant results (**b**) Venn diagrams of differentially expressed genes that overlap between three cucumber transgenic lines (212, 224, and 225) versus the wild-type B10 line. (**c**) Heat map of differentially expressed genes in the fruits of the B10 control and transgenic 212, 224, and 225 lines. DEG expression levels are indicated at the top of the heat map; green indicates down-regulated and red indicates up-regulated genes.

**Figure 3 genes-11-00334-f003:**
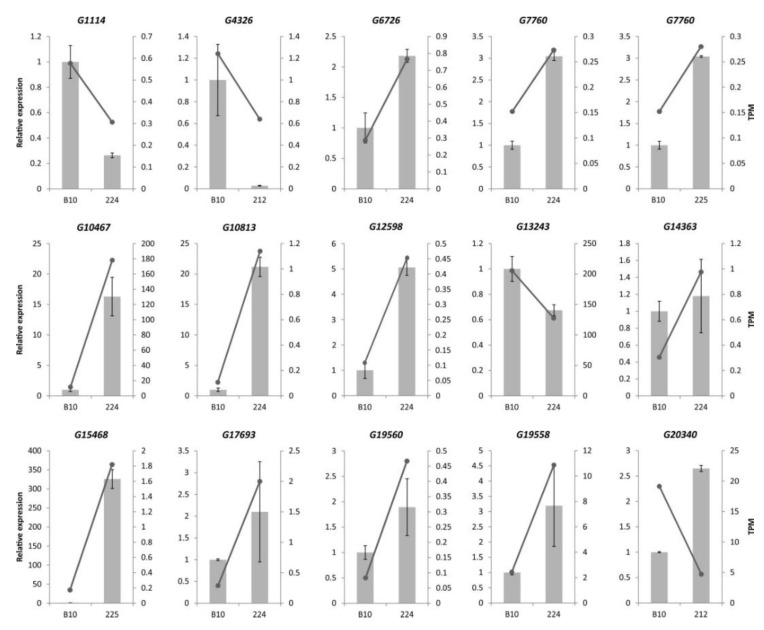
Validation of differentially expressed genes (DEGs) by qPCR. The qPCR results are presented (with grey bars) as relative expression levels. The RNA-seq data (black line) are presented in transcripts per million (TPM).

**Figure 4 genes-11-00334-f004:**
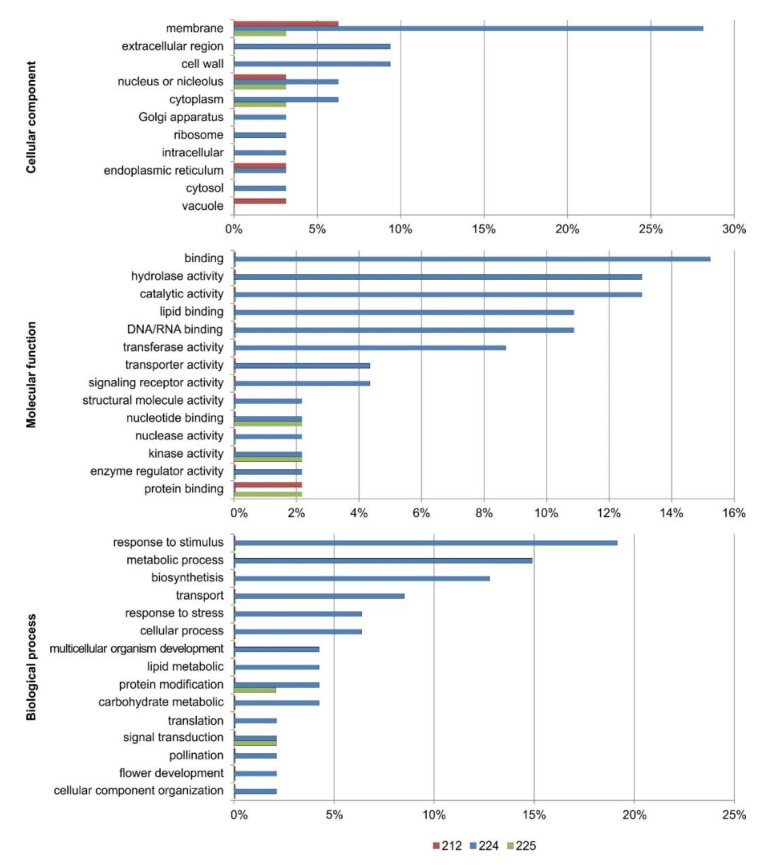
Percentages of GO terms assigned to different functional groups under biological process, cellular component and molecular function categories.

**Figure 5 genes-11-00334-f005:**
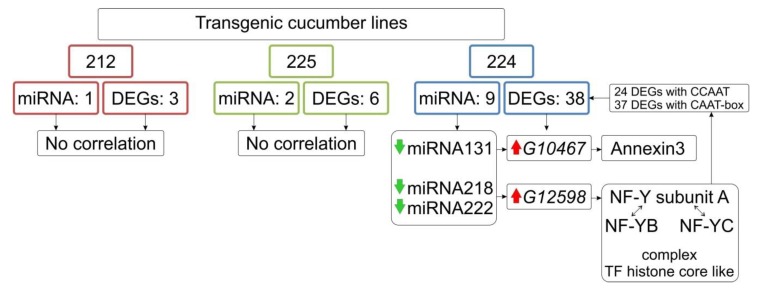
Possible mechanism of epitranscriptome regulation in cucumber transgenic lines.

**Figure 6 genes-11-00334-f006:**
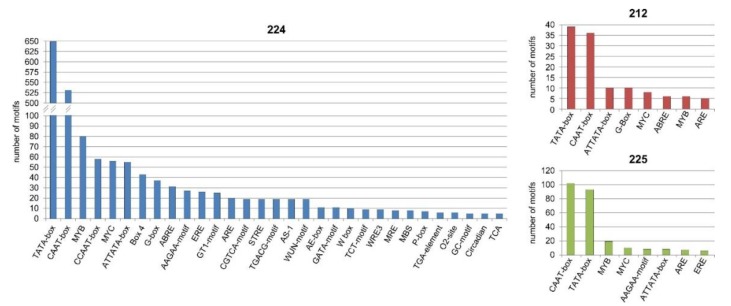
Promoters analysis—occurrence of the different types of motifs in upstream promoter regions of the differentially expressed genes.

**Figure 7 genes-11-00334-f007:**
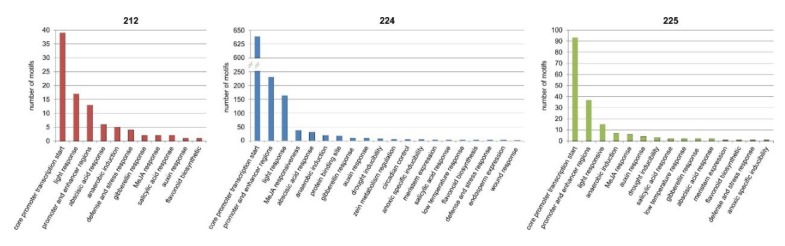
Promoters analysis—occurrence of functional groups of predicted motifs in upstream promoter regions of the differentially expressed genes.

**Table 1 genes-11-00334-t001:** Differentially expressed genes (DEGs) in transgenic lines (212, 224, and 225) versus the wild-type B10 line.

No	Line	Expression	Gene ID	Description	Fold Change	FDR
1	212	down	*G20340*	lincRNA	4.67	6.74 × 10^−6^
2	212	down	*G4326*	PHB domain membrane associated	4.49	1.03 × 10^−8^
3	212	up	*G14398*	Uncharacterized protein	88.90	1.78 × 10^−11^
4	224	down	*G6838*	Restriction endonuclease, type II	9.51	1.03 × 10^−12^
5	224	down	*G16954*	Uncharacterized protein	4.57	5.92 × 10^−12^
6	224	down	*G17313*	Xyloglucan endo-transglycosylase	2.60	5.40 × 10^−4^
7	224	down	*G15599*	thylakoid lumenal 17.4 kDa protein	2.42	9.97 × 10^−4^
8	224	down	*G12465*	Serine threonine-protein kinase	2.38	1,07 × 10^−4^
9	224	down	*G17428*	Uncharacterized protein	2.23	3.87 × 10^−5^
10	224	down	*G11114*	Chaperone DnaJ-domain containing protein	1.87	5.28 × 10^−4^
11	224	down	*G13243*	cystathionine gamma-synthase	1.82	6.27 × 10^−4^
12	224	up	*G853*	lincRNA	36.75	4.42 × 10^−4^
13	224	up	*G21621*	lipase immunity	19.19	1.07 × 10^−4^
14	224	up	*G14592*	ribosomal protein L19	15.91	3.52 × 10^−5^
15	224	up	*G9998*	glycosylphosphatidylinositol-anchored lipid	14.39	3.70 × 10^−4^
16	224	up	*G6441*	plastocyanin-like domain	13.95	1.14 × 10^−4^
17	224	up	*G10686*	hypothetical protein	13.68	8.72 × 10^−4^
18	224	up	*G10813*	protein TRANSPARENT TESTA 12-like	13.48	3.87 × 10^−5^
19	224	up	*G14791*	GDSL esterase lipase	13.28	3.04 × 10^−4^
20	224	up	*G149*	germacrene D synthase-like	12.93	1.68 × 10^−5^
21	224	up	*G17693*	chalcone synthase	11.44	6.66 × 10^−6^
22	224	up	*G3896*	Pollen proteins Ole e I like	10.88	1.22 × 10^−5^
23	224	up	*G7198*	auxin-binding protein ABP19a	10.26	5.17 × 10^−5^
24	224	up	*G12174*	non-specific lipid-transfer proteins	10.17	1.68 × 10^−5^
25	224	up	*G19560*	beta-Amyrin	9.21	6.33 × 10^−4^
26	224	up	*G12648*	Serine hydrolase (FSH1)	8.24	1.28 × 10^−4^
27	224	up	*G8549*	ribonuclease 1 isoform X1	7.60	2.14 × 10^−4^
28	224	up	*G1149*	lincRNA	7.24	3.74 × 10^−4^
29	224	up	*G10467*	annexin D3-like	6.73	2.97 × 10^−4^
30	224	up	*G3425*	AP-2 transcritpion factor	5.19	5.17 × 10^−5^
31	224	up	*G1817*	mannitol dehydrogenase-like	4.83	5.45 × 10^−6^
32	224	up	*G19558*	Acyl carrier protein 4	3.75	7.48 × 10^−5^
33	224	up	*G6726*	Plasma membrane sugar-proton symporter	3.75	8.62 × 10^−4^
34	224	up	*G12598*	Nuclear transcription factor Y subunit	3.55	1.07 × 10^−4^
35	224	up	*G14363*	NADH-ubiquinone oxidoreductase	3.44	3.87 × 10^−5^
36	224	up	*G828*	trehalose 6-phosphate to	3.22	8.72 × 10^−4^
37	224	up	*G9940*	MLP-like protein	2.65	4.95 × 10^−4^
38	224	up	*G6936*	trihelix transcription factor GT	2.49	2.60 × 10^−9^
39	224	up	*G15513*	O-glucosyltransferase 1-like	2.37	1.68 × 10^−5^
40	224	up	*G18456*	Uncharacterized protein	2.01	3.91 × 10^−4^
41	224	up	*G7760*	pentatricopeptide repeat-containing	1.83	1.15 × 10^−4^
42	225	up	*G10371*	lincRNA	104.23	2.71 × 10^−29^
43	225	up	*G14398*	Uncharacterized protein	34.41	5.89 × 10^−8^
44	225	up	*G15468*	lincRNA	19.74	1.15 × 10^−9^
45	225	up	*G6182*	calcium calmodulin-dependent serine	7.73	8.87 × 10^−9^
46	225	up	*G20381*	lincRNA	5.92	3.02 × 10^−9^
47	225	up	*G7760*	pentatricopeptide repeat-containing protein	1.81	4.48 × 10^−4^

**Table 2 genes-11-00334-t002:** Differentially expressed miRNAs in transgenic lines (212, 224, and 225). ^1^ G10467 (ion binding-calcium, annexin), ^2^ G12598 (Nuclear transcription factor Y subunit A).

Line	miRNA	Type of Expression	Log2FC	*p*-Val	No of Targets	Targets in DEGs	Type of Changes
212	320	down	−2.09	5.24 × 10^−5^	20	no	none
224	27	down	−2.02	2.27 × 10^−5^	15	no	none
224	42	up	2.08	6.23 × 10^−6^	22	no	none
224	93	up	2.30	8.70 × 10^−5^	21	no	none
224	123	up	3.55	3.70 × 10^−11^	25	no	none
224	131	down	−2.84	1.11 × 10^−5^	24	*G10467* ^1^	up
224	206	up	2.23	2.69 × 10^−4^	18	no	none
224	218	down	−3.39	4.52 ×10^−7^	28	*G12598* ^2^	up
224	222	down	−3.72	4.96 × 10^−7^	29	*G12598* ^2^	up
224	289	up	3.28	2.45 × 10^−9^	23	no	none
225	206	up	2.09	8.66 × 10^−5^	18	no	none
225	218	down	−2.20	1.42 × 10^−5^	28	no	none

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
