# Peer review of "Effect of Transgenesis on mRNA and miRNA Profiles in Cucumber Fruits Expressing Thaumatin II"

_genes, 2020, doi:10.3390/genes11030334_

Round 1
Reviewer 1 Report
Title: Effect of transgenesis on mRNA and miRNA profiles in cucumber plants expressing thaumatin II
In this work, authors aimed to compare fruits transcriptomes and miRNomes of three cucumber lines (212, 224 and 225), containing the thaumatin II gene, that were obtained by Agrobacterium transformation, with those of a control cucumber line (B10) that I suppose (and hope) was grown in the same environmental conditions as the studied lines.
Authors reported 47 differentially expressed genes (DEGs) between control and the three transgenic lines and 12 differentially regulated miRNAs (three of them that can influence the two targets, that were assigned as DEGs in one of the studied transgenic lines- line 224).
Line 224 was the one showing a higher number of differentially expressed genes which can be explained by the regulation of miR218 and miR222 that caused an up-regulation of the G12598. G12598 codifies for a transcription factor and because of that can have a significant impact on the expression of other genes. Authors also show how the results obtained for this line can explain its increased tolerance to downy mildew.
Authors state that the transformation of cucumber with thaumatin II had minimal impact on gene expression and epigenetics regulation by miRNA in the cucumber fruits.
I think this manuscript present some important data, however in my opinion it may be improved before publication in “Genes”.
Title
“Effect of transgenesis on mRNA and miRNA profiles in cucumber plants expressing thaumatin II”
Reviewer: I would say “cucumber fruits” instead of “cucumber plants”. This study was performed in the fruits.
Abstract
Lines 20-21 “we also identified (…) can influence the two targets (assigned as DEGs) in line 224”
Reviewer: At this stage of the manuscript readers don’t know which line is this, I would say “(…) in one of the studied transgenic lines (line 224).”
Lines 23-25: “Moreover, the phenotypic traits …”
Reviewer: Which “phenotypic traits” as far as I could understand they were not reported in this study.
Materials and methods
Lines 85-86 : “ Three transgenic cucumber lines (212, 224, and 225) (…) analysis.”
Reviewer: I would say: “Four cucumber lines, three transgenic lines containing thaumatin II gene (212, 224, 225) and one control line (B10), were used for the RNA sequencing (RNA-seq) analysis”
Lines 89-91: “To develop lines (…) under specific strictly controlled and specific greenhouse conditions…”
Reviewer: This is not scientific. Which conditions were those? Authors must specify which were the used growth conditions.
Reviewer: Also very important in this item:
Which generation was analyzed. As far as I could understand, authors have grown control and transgenic lines under controlled greenhouse conditions for 9 generations and then used seeds from F9 generation that were grown in the field. Right?
So fruits from F10 were analyzed, right? Please clarify this.
It is not clear if B10 line had also passed through all the generations in the exact same environmental conditions as transgenic lines. I hope that this have happened...this fact is essential for the experiment. Otherwise, differences in environmental growth conditions may per se justify the found transcriptome and miRNomes differences…
Results
Figure 2
Reviewer: On figure 2a red dots are not much visible and I suppose they will not be visible for color blind people. I would suggest, if possible, to use different shapes.
On Figure 2c DEGs identifications are extremely small.
3.4. DEGs in the transgenic 224 line
Lines 233 :”the function of two of the genes was uncharacterized (G16954 and G17428)”
and line 240: “one has no function (G3896)”
Reviewer: so here you can say that "the function of three of the genes was uncharacterized (G16954, G17428 and G3896)"
3.6. Confirmation of Illumina RNA-seq expression by qPCR
Lines 252-253: “To validate the DEG expression (…)16 DEGs for verification…”
Reviewer: I suppose you mean “15 DEGs” (G20340, G4326, G17693, G13243, G10813, G10467, G12598, G6726, G19560, G1114, G14363, G19558, G7760 (224), G7760 (125) G15468).
Lines 256-259: The gene expression levels (…)were positively correlated for 15 (94%) (…) of the two methods and algorithms.”
Reviewer: I suppose you mean “for 14 (93%)…”
Figure 3
Reviewer: Pay attention to the label “TPM” in this figure, some of them are overlaid to the numbers of the axis.
3.7. Functional annotation of the DEGs
Line 274: “and biosynthetic processes in theb224 line”
Reviewer: Please substitute “theb224 line” by “the 224 line”
Line 279: “and in 225 membranes, nucleus and cytoplasm in 225 line…”
Reviewer: Substitute by “and membranes, nucleus and cytoplasm in 225 line…”
Figure 5
Upper right text box: “12 DEGs with CCAAT”
Reviewer: I suppose you mean 24 DEGs with CCAAT, right?
Discussion
Line 466: “detected such kind of 24 genes”
Reviewer: I suppose you mean “detect such kind of motif in 24 genes”
Author Response
Reviewer 1
Comments and Suggestions for Authors
Title: Effect of transgenesis on mRNA and miRNA profiles in cucumber plants expressing thaumatin II
In this work, authors aimed to compare fruits transcriptomes and miRNomes of three cucumber lines (212, 224 and 225), containing the thaumatin II gene, that were obtained by Agrobacterium transformation, with those of a control cucumber line (B10) that I suppose (and hope) was grown in the same environmental conditions as the studied lines.
Response: Yes, control and transgenic lines were grown in the same environmental conditions, and were collected at the same time point and further proceedings were also common to minimize the impact of external factors on the variability of materials.
Authors reported 47 differentially expressed genes (DEGs) between control and the three transgenic lines and 12 differentially regulated miRNAs (three of them that can influence the two targets, that were assigned as DEGs in one of the studied transgenic lines- line 224). Line 224 was the one showing a higher number of differentially expressed genes which can be explained by the regulation of miR218 and miR222 that caused an up-regulation of the G12598. G12598 codifies for a transcription factor and because of that can have a significant impact on the expression of other genes. Authors also show how the results obtained for this line can explain its increased tolerance to downy mildew. Authors state that the transformation of cucumber with thaumatin II had minimal impact on gene expression and epigenetics regulation by miRNA in the cucumber fruits. I think this manuscript presents some important data, however in my opinion it may be improved before publication in “Genes”.
Response: Thank you so much for paying attention to what we should refine. We have referred to all comments posted below, correcting the manuscript and hopefully as expected. Thank you that the reviewer pointed out the importance of the conducted research.
Title
“Effect of transgenesis on mRNA and miRNA profiles in cucumber plants expressing thaumatin II”
Reviewer: I would say “cucumber fruits” instead of “cucumber plants”. This study was performed in the fruits.
Response: The amendment to the title proposed by the reviewer is definitely more appropriate, the title has been changed (Line 3).
Abstract
Lines 20-21 “we also identified (…) can influence the two targets (assigned as DEGs) in line 224”
Reviewer: At this stage of the manuscript readers don’t know which line is this, I would say “(…) in one of the studied transgenic lines (line 224).”
Response: Yes, indeed this can be misleading. We have changed sentences as suggested (Line 22).
Lines 23-25: “Moreover, the phenotypic traits …”
Reviewer: Which “phenotypic traits” as far as I could understand they were not reported in this study.
Response: Thank you for that comment. We deleted this sentence from the abstract (Line 24 - 25 deleted).
Materials and methods
Lines 85-86 : “ Three transgenic cucumber lines (212, 224, and 225) (…) analysis.”
Reviewer: I would say: “Four cucumber lines, three transgenic lines containing thaumatin II gene (212, 224, 225) and one control line (B10), were used for the RNA sequencing (RNA-seq) analysis”
Response: This sentence was corrected according to the reviewer's comments (Line 86-87).
Lines 89-91: “To develop lines (…) under specific strictly controlled and specific greenhouse conditions…”
Reviewer: This is not scientific. Which conditions were those? Authors must specify which were the used growth conditions.
Response: Plants were cultivated in 16 h photoperiod (light intensity 1500 μmol·m-2·s-1) with temperature 25°C-27°C during the day and 18°C-20° during the night as it is described in Szwacka et al. 2002. This sentence was added in the text (Line 93).
Reviewer: Also very important in this item:
Which generation was analyzed. As far as I could understand, authors have grown control and transgenic lines under controlled greenhouse conditions for 9 generations and then used seeds from F9 generation that were grown in the field. Right?
So fruits from F10 were analyzed, right? Please clarify this.
Response: Since receiving the primary transformants described in 2002, on average every 2 years we performed self-pollination under control conditions in the greenhouse. Almost every generation was further characterized and also were described as homozygous (Szwacka 2009). Currently, we were at the T9 generation. In this work the seeds of the T9 generation were sown in the greenhouse and then transferred to the field and the pulp from the fruit of this generation was used for analysis in this work. We did not analyze the F10 generation, because we obtained tissue from slices of pulp without seeds (F10). This was inaccurately described at work. We made the appropriate corrections (Line 93).
It is not clear if B10 line had also passed through all the generations in the exact same environmental conditions as transgenic lines. I hope that this have happened...this fact is essential for the experiment. Otherwise, differences in environmental growth conditions may per se justify the found transcriptome and miRNomes differences…
Response: For the transformation experiment a stably homozygous B10 line was taken. As we showed in a previous publication (Osipowski et al 2020) there are very little changes between generations of B10 lines. Reproduction of the B10 line has always been carried out in a greenhouse to control the conditions and minimize the effect of insects. Cucumbers were pollinated manually by qualified staff. The cultivation of transgenic cucumbers along with the B10 line has always been carried out in a greenhouse in the same environmental conditions. Transgenic cucumbers must not be sown in the field at all in Poland. Only for the purposes of this and selected previous projects we obtained the consent of the Minister of Environment for the release of transgenic plants to field conditions to obtain production conditions (Paragraph 2.1 Lines 90 - 99).
Results
Figure 2
Reviewer: On figure 2a red dots are not much visible and I suppose they will not be visible for color blind people. I would suggest, if possible, to use different shapes.
On Figure 2c DEGs identifications are extremely small.
Response: Thank you for that comment. We modified Figure 2a by drawing bigger red and green asterisks, we also increased the size of the font on Figure 2c (Line 223, 225).
3.4. DEGs in the transgenic 224 line
Lines 233 :”the function of two of the genes was uncharacterized (G16954 and G17428)”
and line 240: “one has no function (G3896)”
Reviewer: so here you can say that "the function of three of the genes was uncharacterized (G16954, G17428 and G3896)"
Response: Thank you for that comment according to the Reviewer's comment we modified the sentence (Lines 239, 246).
3.6. Confirmation of Illumina RNA-seq expression by qPCR
Lines 252-253: “To validate the DEG expression (…)16 DEGs for verification…”
Reviewer: I suppose you mean “15 DEGs” (G20340, G4326, G17693, G13243, G10813, G10467, G12598, G6726, G19560, G1114, G14363, G19558, G7760 (224), G7760 (125) G15468).
Response: Thank you for that comment. We put the wrong number of validated DEGs by mistake, we corrected it in the text (lines 259, 260).
Lines 256-259: The gene expression levels (…)were positively correlated for 15 (94%) (…) of the two methods and algorithms.”
Reviewer: I suppose you mean “for 14 (93%)…”
Response: Thank you for that comment. As explained above, we put the wrong number of validated DEGs by mistake and therefore the calculation of positive correlation was also wrong. We have changed that in the manuscript (Line 263).
Figure 3
Reviewer: Pay attention to the label “TPM” in this figure, some of them are overlaid to the numbers of the axis.
Response: Thank you, we changed Figure 3 to make it more readable (Line 267).
3.7. Functional annotation of the DEGs
Line 274: “and biosynthetic processes in theb224 line”
Reviewer: Please substitute “theb224 line” by “the 224 line”
Response: Thank you for pointing this out, we did not notice this typo (Line 280).
Line 279: “and in 225 membranes, nucleus and cytoplasm in 225 line…”
Reviewer: Substitute by “and membranes, nucleus and cytoplasm in 225 line…”
Response: Thank you for this comment, as we did not notice this duplication. We made corrections in the text (Line 285).
Figure 5
Upper right text box: “12 DEGs with CCAAT”
Reviewer: I suppose you mean 24 DEGs with CCAAT, right?
Response: Thank you for pointing this mistake. Yes, it should be 24, and the Figure 5 is corrected (Line 306).
Discussion
Line 466: “detected such kind of 24 genes”
Reviewer: I suppose you mean “detect such kind of motif in 24 genes”
Response: Thank you, indeed the sentence is misleading, so we corrected it according to suggestion (Line 455).
Reviewer 2 Report
Original scientific paper, solves in detail and precisely the issue of considerable practical importance.
The paper is a continuation of the research issue addressed by the team of authors, which is also an important contribution in terms of obtaining significant results.
As regards the methodology of the experiment, the following data would need to be added:
- What criteria the authors considered when selecting the tested transgenic lines (212, 224 and 225)? There are missing the characteristics of testes lines.
- Was the stability of transgene expression checked in every next generation after self-pollination?
- What was the reason to select for the analyses just T9 generation?
- In the abstract the authors state that “the phenotypic traits of the transgenic cucumber plants were the same as those of the non-transgenic wild-type plants”. However, a closer specification of these phenotypic traits is missing.
The obtained results are described in detail and the factual and professional discussion points to the professional qualifications of the authors and the precision of their work.
However, the manuscript lacks a detailed explanation of the divisional response of individual genomes of transgenic lines in terms of gene expression and miRNAs activity. How authors would explain the highest number of DEGs in the line 224? As one explanation, they assume an interaction between the target sequence of miR218 and miR222. However, this type of regulation would need to be verified.
The authors conclude that ... “transformation using A. tumefaciens generated a relatively small number of unintentional changes...” Describing the results we can see that most of the differentially expressed genes in the line 224 (where it was recorded the highest number of DEGs) were related to plant stress or defence responses. How the authors explain the relationship between thaumatin II introduction and plant genomic response induction in the context of resistance?
In the section 3.8 is incorrect information provided. In the line 224, based on the Table 2, five differentially expressed miRNAS was up-regulated, not four. Similarly in the section between rows 447 and 448 provides incorrect information... “All miRNAs were down-regulated in the 224 line”... only four of them (Table 2).
It would be appropriate for authors to define the conclusions of their research more specifically. Opinion, that the recorded changes in miRNAs activity in transgenic lines were minimal, I would define a little more carefully, as the authors have so far characterized the type of expression of differentially expressed miRNAs, not their target sequences.
The manuscript requires minor formal modifications:
-row 35; Thaumatococcus daniellii (Benn.) Benth.
-row 61; Agrobacterium tumefaciens
-row 93; ... and phenotypically assessed. (there are 2 dots)
-row 123; 72°C for 10 min (missing space)
-row 228; ... analysis indicated that the ....
-rows 345 and 346; DnaJ gene
-row 366; thaumatin II.
-row 374; pathogen Botrytis sp.
-row 447; The type...
Author Response
Reviewer 2
Comments and Suggestions for Authors
Original scientific paper, solves in detail and precisely the issue of considerable practical importance.The paper is a continuation of the research issue addressed by the team of authors, which is also an important contribution in terms of obtaining significant results.
Response: We would like to thank the Reviewer for noticing the importance of our research. We are pleased that our work and its practical and developmental significance were appreciated.
As regards the methodology of the experiment, the following data would need to be added:
- What criteria the authors considered when selecting the tested transgenic lines (212, 224 and 225)? There are missing the characteristics of testes lines.
Response: Since we received the primary transformants described in 2002, we conducted self-pollination every two years under control conditions in a greenhouse. Almost every generation has been further characterized and also described as homozygous (Szwacka 2009, 2012). In the tests presented in previous publications concerning cucumber transgenic lines, these lines showed the most stable expression.
- Was the stability of transgene expression checked in every next generation after self-pollination?
Response:Stability of transgene expression has been tested in each subsequent generation after self-pollination, and these lines were considered as homozygous (Szwacka et al. 2009, 2012). In this work we also confirmed the presence and expression of transgene (Line 204, Fig 1).
- What was the reason to select for the analyses just T9 generation?
Response: As mentioned above transgenic lines were self-pollinated on average every 2 years. Almost every generation was further characterized. Currently, we were at the T9 generation. In this work the seeds of the T9 generation were sown in the greenhouse and then transferred to the field and the pulp from the fruit of this generation was used for analysis in this work (Line 93).
In the abstract the authors state that “the phenotypic traits of the transgenic cucumber plants were the same as those of the non-transgenic wild-type plants”. However, a closer specification of these phenotypic traits is missing.
Response: Thank you for that comment. The phenotypic traits were the same in control and transgenic lines. We have photo-documentation of the phenotypes, but we did not want to present these figures as it was not the aim of this paper and it is not important to present it. We deleted this sentence from the abstract (Line 24 - 25).
The obtained results are described in detail and the factual and professional discussion points to the professional qualifications of the authors and the precision of their work.
Response: Thank you very much that the Reviewer noted that the results are described in detail, and the substantive and professional discussion indicates the professional qualifications of the authors and the precision of their work. We feel extremely appreciated.
However, the manuscript lacks a detailed explanation of the divisional response of individual genomes of transgenic lines in terms of gene expression and miRNAs activity. How authors would explain the highest number of DEGs in the line 224? As one explanation, they assume an interaction between the target sequence of miR218 and miR222. However, this type of regulation would need to be verified.
Response: Based on the data presented in this work, we cannot draw conclusions about what is happening in the genomes of transgenic lines. It is known that the site of transgene integration has a huge impact on the expression of new genes as well as it can directly and indirectly affect the expression of endogenous genes that are in the plant. It seemed to us that speculation in this direction goes too far beyond what we present. The claims would not be substantiated. However, we performed sequencing of the genomes of transgenic lines and currently we are examining the sites of transgene integration. These results will be present in our next publication. Referring to the results presented in this paper, we can discuss the impact of miRNA on targets and the regulation of other genes for now. Due to the relationships found in this work (Line 302 Fig 5), we decided to continue this thread in the future to verify these adjustment paths. At the moment we show the probable correlation we have noticed from RNA-seq and miRNA-seq data, and validated targets through the use of qPCR (Line 267 Fig 3).
The authors conclude that ... “transformation using A. tumefaciens generated a relatively small number of unintentional changes...” Describing the results we can see that most of the differentially expressed genes in the line 224 (where it was recorded the highest number of DEGs) were related to plant stress or defence responses. How the authors explain the relationship between thaumatin II introduction and plant genomic response induction in the context of resistance?
Response: Thaumatin are pathogenesis-related (PR) proteins that are induced by many factors, from hormones (in example ethylene) to pathogens, and are structurally diverse and ubiquitous in plants (Ruiz-Medrano et al 1992). Proteins (including thaumatin, osmotin and major and minor PR proteins) are involved in systemic acquired immunity and stress response in plants, although their exact role is unknown. In our work we try to point out the function of DEGs, which (as discussed in manuscript) could be correlated with a process of higher immune tolerance against pathogens. Additionally we identified relatively small amounts of DEGs (from 3 to 37) in the transgenic lines. We are currently conducting research in the area of transcriptome comparison of genes expression in chemical mutant lines and lines presenting somaclonal variation. It turns out that in somaclonal and mutants lines the number of DEGs is much higher than in transgenic lines with thaumatin, hence our statement of a relatively small number of changes. Nevertheless, we decided to remove this sentence (Line 495 - 497).
In the section 3.8 is incorrect information provided. In the line 224, based on the Table 2, five differentially expressed miRNAS was up-regulated, not four. Similarly in the section between rows 447 and 448 provides incorrect information... “All miRNAs were down-regulated in the 224 line”... only four of them (Table 2).
Response: Thank you for pointing out this mistake. We corrected this in the text (Line 296-298, 455).
It would be appropriate for authors to define the conclusions of their research more specifically. Opinion, that the recorded changes in miRNAs activity in transgenic lines were minimal, I would define a little more carefully, as the authors have so far characterized the type of expression of differentially expressed miRNAs, not their target sequences.
Response: Indeed, we have presented the conclusions on miRNA activity too generally. As suggested by the reviewer, we redefined the conclusion of our research to be more specific because it relates to line 224. In this line, two miRNA targets were checked by qPCR analysis and different expression was confirmed. The information of the changes in target expression and verification by qPCR has been added in the text in the discussion part (Line 457).
The manuscript requires minor formal modifications:
-row 35; Thaumatococcus daniellii (Benn.) Benth.
-row 61; Agrobacterium tumefaciens
-row 93; ... and phenotypically assessed. (there are 2 dots)
-row 123; 72°C for 10 min (missing space)
-row 228; ... analysis indicated that the ....
-rows 345 and 346; DnaJ gene
-row 366; thaumatin II.
-row 374; pathogen Botrytis sp.
-row 447; The type...
Response: All of the above are corrected according to the reviewer's comments.